# Machine learning-based pipeline for automated intracerebral hemorrhage and drain detection, quantification, and classification in non-enhanced CT images (NeuroDrAIn)

**Samer Elsheikh** [1]*, **Ahmed Elbaz**[1], **Alexander Rau**[1], **Theo Demerath**[1], **Elias Kellner**[2], **Ralf Watzlawick**[3], **Urs Würtemberger**[1], **Horst Urbach**[1], **Marco Reisert**[2,4]

1 Department of Neuroradiology, Medical Center–University of Freiburg, Faculty of Medicine, University of Freiburg, Freiburg, Germany, 2 Medical Physics, Department of Diagnostic and Interventional Radiology, Medical Center, Faculty of Medicine, University of Freiburg, Freiburg, Germany, 3 Department of Neurosurgery, Medical Center-University of Freiburg, Faculty of Medicine, University of Freiburg, Freiburg, Germany, 4 Department of Stereotactic and Functional Neurosurgery, Medical Center—University of Freiburg, Faculty of Medicine, University of Freiburg, Freiburg, Germany

* samer.elsheikh@uniklinik-freiburg.de

## Abstract

### Background and purpose

External drainage represents a well-established treatment option for acute intracerebral hemorrhage. The current standard of practice includes post-operative computer tomography imaging, which is subjectively evaluated. The implementation of an objective, automated evaluation of postoperative studies may enhance diagnostic accuracy and facilitate the scaling of research projects. The objective is to develop and validate a fully automated pipeline for intracerebral hemorrhage and drain detection, quantification of intracerebral hemorrhage coverage, and detection of malpositioned drains.

### Materials and methods

In this retrospective study, we selected patients (n = 68) suffering from supratentorial intracerebral hemorrhage treated by minimally invasive surgery, from years 2010–2018. These were divided into training (n = 21), validation (n = 3) and testing (n = 44) datasets. Mean age (SD) was 70 (±13.56) years, 32 female. Intracerebral hemorrhage and drains were automatically segmented using a previously published artificial intelligence-based approach. From this, we calculated coverage profiles of the correctly detected drains to quantify the drains' coverage by the intracerebral hemorrhage and classify malpositioning. We used accuracy measures to assess detection and classification results and intraclass correlation coefficient to assess the quantification of the drain coverage by the intracerebral hemorrhage.

**Data Availability Statement:** All relevant data are within the manuscript and its Supporting Information files. Python, MATLAB and R scripts

for segmentation, quantification and classification are available in https://github.com/s-elsheikh/NeuroDrAIn under MIT license.

**Funding:** The author(s) received no specific funding for this work.

**Competing interests:** Unrelated: research grants from Bracco Suisse S.A., Medtronic. Travel grant from Medtronic, received honoraria for lectures from Penumbra. Horst Urbach: Received honoraria for lectures from Biogen, Eisai, Mbits and Lilly, is supported by German Federal Ministry of Education and Research, and is coeditor of Clin Neuroradiol. Elias Kellner: Shareholder of and received fees from VEObrain GmbH, Freiburg, Germany. Theo Demerath: No competing interest (unrelated: travel grants Balt, Stryker). We confirm that this does not alter our adherence to PLOS ONE policies on sharing data and materials.

## Results

In the test dataset, the pipeline showed a drain detection accuracy of 0.97 (95% CI: 0.92 to 0.99), an agreement between predicted and ground truth coverage profiles of 0.86 (95% CI: 0.85 to 0.87) and a drain position classification accuracy of 0.88 (95% CI: 0.77 to 0.95) resulting in area under the receiver operating characteristic curve of 0.92 (95% CI: 0.85 to 0.99).

## Conclusion

We developed and statistically validated an automated pipeline for evaluating computed tomography scans after minimally invasive surgery for intracerebral hemorrhage. The algorithm reliably detects drains, quantifies drain coverage by the hemorrhage, and uses machine learning to detect malpositioned drains. This pipeline has the potential to impact the daily clinical workload, as well as to facilitate the scaling of data collection for future research into intracerebral hemorrhage and other diseases.

## Introduction

Intracerebral hemorrhage (ICH) constitutes a significant cause of morbidity and mortality, with an estimated global incidence of about 5 million events per year [1]. To reduce the mass effect of ICH, evacuation of the clot is feasible by positioning a drain in the bleeding using minimally invasive surgery (MIS). Current guidelines recommend MIS in patients with supratentorial ICH > 20–30 ml volume and Glasgow coma scale of 5–12 [2]. Until recently, randomized trials could not prove the efficacy of MIS [3]. However, the recently published ENRICH trial confirmed the superiority of MIS over conservative management [4, 5]. The methods employed in the MISTIE III trial, the largest published study on MIS for ICH to date, involved a CT scan after surgery to confirm correct placement of the drain, and repositioning of the catheter in cases of insufficient placement [3].

Multiple techniques were introduced for the placement of drains in acute ICH; some of these rely on image or augmented reality assistance, which can improve precision but also increases time and cost of the procedure [6]. They also noted that the freehand technique is still commonly employed for drain placement, resulting in frequent suboptimal positions. Moreover, the lack of research on standardized or clinically validated criteria for detecting malpositioned drains represents a significant limitation to both clinical and scientific evaluation. Consequently, the assessment of these images remains highly subjective. The application of automated segmentation techniques using convolutional neural networks to medical images provides the potential for the generation of quantitative data. If validated, this data could facilitate objective clinical assessment and serve as a foundation for further scientific research.

In this study, we assessed the potential of an end-to-end pipeline for the evaluation of ICH drain positioning after MIS in acute supratentorial ICH. This pipeline relies on a previously developed ICH and drain segmentation model [7] to accurately detect drains, automate quantification of drain coverage by the ICH and artificial intelligence-based classification of malpositioned drains.

## Materials and methods

The retrospective study was approved by the ethics committee of the Albert-Ludwigs-Universität in Freiburg. Obtaining informed written or verbal consent was waived. We aimed to develop a pipeline (Fig 1) including:

**Fig 1. Chart delineating the inputs and outputs of each step of the pipeline.** 1: Detection of true positive drain objects in binary mask, 2: quantification of the coverage profile and 3: classification of the drain position.

1. Detection of the drains as discrete objects.

2. Quantification of the drain coverage by the ICH.

3. Classification of the drain position.

The same cohort was used to develop a convolutional neural network for ICH and drain segmentation [7]. Initially, 29 patients ≥ 18 years of age, who suffered from supratentorial ICH and were treated with MIS between 2011 and 2018 were randomly selected from our imaging archiving system. These were partitioned into training (n = 21, 29 scans), validation (n = 3, 4 scans) and testing (n = 5, 6 scans) datasets. To avoid bias in our results, another 39 consecutive patients (53 scans) examined between 2010 and 2012 were added to the testing dataset, resulting in a total number of n = 44 patients in the independent testing dataset (59 scans). To avoid data leakage, patients already selected in one of the other groups were excluded (n = 13) and all scans belonging to the same patient were assigned to the same group. To further address potential sources of bias, no exclusions were made based on scanner model, scanning parameters, voxel size, or image quality. Image selection was performed on June 21, 2023. Patient and imaging data was pseudonymized before analysis.

In the present study, we employed the segmentation results of a previously published model [7] to quantify the coverage of the correctly detected drains by the ICH and identify malpositioned drains. The model was developed using the Patchwork CNN Toolbox, as previously described [8]. The model architecture employs hierarchical patching approach to address challenges posed by large segmentation tasks, such as those commonly encountered in medical imaging, while anatomical information. The published model employed 3 scales; the finest scale was reformatted to 1-mm isotropic voxels. Within each scale, a Unet-inspired architecture was applied, using leaky RELU as the activation function. The training loss was evaluated using the binary cross-entropy loss function. Six distinct model variations were created to tune different hyperparameters; feature dimensions at each hierarchical scale, loss function, and augmentation parameters (random rotation, zoom, and flip). Validation was conducted using a holdout dataset. This approach may have reduced the robustness of the model, but it also mitigated the higher computational cost of a cross-validation approach. Notwithstanding the small size of the training dataset, a marginal difference between the results in the training and validation datasets was observed, suggesting minimal or no overfitting. For model evaluation, spatial metrics were reported [9]. Although surface measures are considered superior for evaluating 3D segmentation models, the 2D dice similarity coefficient was selected to facilitate comparison with other published articles on ICH segmentation. In the final test group, the published model achieved a dice similarity coefficient of 0.86 and 0.91 and a surface dice similarity coefficient of 0.79 and 0.95 in the segmentation of ICH and drains, respectively [7].

## Ground truth (GT)

Three neuroradiologists (TD, SE, AR), with 8, 18 and 5 years of experience, respectively, independently evaluated the drain position on a local instance of the Nora imaging platform (https://www.nora-imaging.com). Only the distal 15 mm of the drain were taken into account, as the openings are located in this segment (S1 Fig). External ventricular drains were also included in the evaluation. To avoid ambiguity, a numerical identifier was added to each drain if more than one drain was present within a single patient. Correct drain position was assessed using a binary scale ("correct" or "not correct"). Measurement and 3D reformatting tools were available to the readers. Discrepancies were resolved by majority agreement.

## Drain detection

Utilizing the drain probability masks of the previously published model [7], we calculated accuracy measures of drain detection at multiple probability thresholds (range = 0.5–0.99, steps = 0.01). We used GT masks to label discrete objects as either "drain" or "noise".

## Quantification of the drain coverage profile

Using the probability masks of the ICH and the correctly detected drains, in MATLAB (MATLAB R2021a, The MathWorks) we extracted the profile of the ICH and drain relative overlap. We first calculated a "volume of touch" (V) between ICH and drain. V was computed via distance transforms: a voxel within the drain volume belonged to V if its spatially closest non-drain voxel belonged to the ICH and not to the background. To establish a normalized coordinate frame, we used the eigensystem of the moment tensor of V, in which the profiles were always along the major axis of the eigensystem. For quantification, images were regridded at 0.5-mm isotropic resolution, followed by Gaussian smoothing to 1-mm voxels. Output was the relative area (orthogonal to the major axis) of V to the total drain area. This process was repeated iteratively on each discrete object in the binary mask.

To optimize the agreement between prediction and GT coverage profiles, we tested multiple probability thresholds in the drain mask (range = 0.05–0.50, steps = 0.01).

## Classification of drain position

We used the predicted coverage profiles of the drains by the ICH for model training. To develop a binary machine learning classification model for the position of the drain in relation to ICH, we tested 10 different model variations, from various model families, including logistic regression, neural networks and random forest. Parameter tuning was performed using leave-one-out cross-validation with a random grid of 60 different parameter combinations to optimize the area under the receiver operating characteristic curve (AUC-ROC). We used the "caret" package version 6.0–94 in R software version 4.2.0 [10, 11]. Independent variables were the normalized (centered and scaled) numeric values of the predicted coverage profile in the distal 15 mm of the drain. The dependent variable was the subjective human GT evaluation of the drain position.

The small size of our data set may present challenges in developing a predictive classification model. To prevent overfitting we employed regularized models, but did not exclude any of the more complex model families. To limit bias we utilized leave-one-out cross validation, which would maximize the amount of data during training. Finally, we tested model robustness on an independent testing dataset. As a thorough evaluation of the performance could prove difficult, we included the 95%-confidence intervals in all our results. Still we recognize that a thorough evaluation of model performance will be difficult.

## Statistics

Statistical evaluation of the results and plotting were done using R software version 4.2.0 [10].

1. Ground Truth reading of drain position: Fleiss' kappa for inter-rater agreement, using the "IRR" package version 0.84.1 [12].

2. Drain detection and classification accuracy: AUC-ROC, accuracy, sensitivity and specificity [11, 13].

3. Drain coverage profile: Two-way random-effects model, single-measure intraclass correlation coefficients (ICC) were computed using the "IRR" package, to assess the agreement between coverage profiles of predicted and GT volumes [12]. We visualized the disagreement using concordance plots and Bland-Altman plots using the "blandr" package version 0.5.1 [14].

# Results

## Dataset characteristics

The mean age (SD) of included cohort was 70 (±13.56) years, 32 female. Table 1 provides further details on the characteristics of patients and drains in each group. Images were acquired from a single center on three different scanners. Voxel sizes encompassed a range of 0.38–0.52 x 0.38–0.52 x 0.7–5 mm$^3$.

## Human readings

In our cohort 36 (37.5%) of the ICH drains, as well as all external ventricular drains (n = 11), were incorrectly positioned. The distribution of "correct" and "not correct" position of the ICH drain tip in each group is shown in Table 2. Overall, a moderate inter-rater agreement (Fleiss's $\kappa$ = 0.57) was observed.

## Drain detection

We observed 100% accuracy over a wide range of probability thresholds in the training and validation datasets. As we intended to avoid false positive results, we used 0.9 as a threshold. Applying this threshold on the testing group, we achieved an accuracy of 0.97 (124 of 128; 95%

**Table 1. Patient and drain characteristics in all datasets.**

| Variable | Training | Validation | Testing |
|---|---|---|---|
| Dataset characteristics | | | |
| No. of Patients | 21 | 3 | 44 |
| No. of CT scans | 29 | 4 | 59 |
| Age | 71.4 (±14.7) | 67.7 (±10.7) | 70 (±13.4) |
| No. of female patients | 14 (66.7%) | 1 (33.3%) | 17 (38.6%) |
| Drain characteristics | | | |
| No. of drains | 36 | 4 | 67 |
| No. of ICH drains | 30 | 4 | 62 |
| No. of EVDs | 6 | 0 | 5 |

Age is presented as mean ±SD. Categorical variables are presented as a number with percentage in parenthesis. ICH: intracerebral hemorrhage. EVDs: external ventricular drains.

**Table 2. Results of each step in all data partitions.**

| Variable | Training | Validation | Testing |
|---|---|---|---|
| Drain position and reader agreement | | | |
| No. of correct ICH drains | 18 (60%) | 1 (25%) | 41 (66.1%) |
| No. of not correct ICH drains | 12 (40%) | 3 (75%) | 21 (33.9%) |
| Fleiss κ | 0.59 | 0.63 | 0.54 |
| Drain detection—baseline | | | |
| No. of drain objects at 0.5 threshold | 36 | 4 | 67 |
| No. of noise objects at 0.5 threshold | 4 | 2 | 61 |
| Threshold range—100% drain detection | 0.61–0.98 | 0.69–0.98 | 0.99–0.99 |
| Drain detection accuracy measures at threshold 0.9 | | | |
| Accuracy (CI) | 1 (95% CI: 0.91 to 1) | 1 (95% CI: 0.54 to 1) | 0.97 (95% CI: 0.92 to 0.99) |
| Sensitivity (CI) | 1 (95% CI: 0.9 to 1) | 1 (95% CI: 0.4 to 1) | 0.97 (95% CI: 0.9 to 1) |
| Specificity (CI) | 1 (95% CI: 0.4 to 1) | 1 (95% CI: 0.16 to 1) | 0.97 (95% CI: 0.89 to 1) |
| Coverage profiles' agreement | | | |
| Probability threshold for optimized ICC | 0.44 | 0.48 | 0.33 |
| ICC at optimized probability threshold (CI) | 0.95 (95% CI: 0.94 to 0.95) | 0.79 (95% CI: 0.74 to 0.84) | 0.87 (95% CI: 0.86 to 0.87) |
| ICC at probability threshold 0.44 (CI) | 0.95 (95% CI: 0.94 to 0.95) | 0.77 (95% CI: 0.72 to 0.82) | 0.86 (95% CI: 0.85 to 0.87) |
| Average disagreement (CI) | -0.014 (95% CI:-0.26 to 0.23) | -0.02 (95% CI:-0.46 to 0.42) | -0.005 (95% CI:-0.36 to 0.35) |
| Disagreement: No. of outliers | 164 of 2196 (7.47%) | 22 of 244 (9.02%) | 305 of 3965 (7.69%) |
| Drain classification accuracy measures | | | |
| Accuracy (CI) | 0.97 (95% CI: 0.85 to 1) | 1 (95% CI: 0.4 to 1) | 0.88 (95% CI: 0.77 to 0.95) |
| Sensitivity (CI) | 0.94 (95% CI: 0.73 to 1) | 1 (95% CI: 0.03 to 1) | 0.9 (95% CI: 0.77 to 0.97) |
| Specificity (CI) | 1 (95% CI: 0.81 to 1) | 1 (95% CI: 0.29 to 1) | 0.83 (95% CI: 0.63 to 0.95) |
| AUC-ROC (CI) | 0.98 (95% CI: 0.93 to 1) | 1 (95% CI: NA to NA) | 0.92 (95% CI: 0.85 to 0.99) |

Categorical variables are presented as a number with percentage in parenthesis. ICH: intracerebral hemorrhage. ICC: Intraclass correlation coefficient. AUC-ROC: area under the receiver operating characteristic curve. NA: not applicable.

CI: 0.92 to 0.99), sensitivity of 0.97 (65 of 67; 95% CI: 0.9 to 1) and a specificity of 0.97 (59 of 61; 95% CI: 0.89 to 1) (Table 2).

We examined the drains that were inaccurately detected (n = 4, S2 Fig):

1. False negative results (n = 2): The pipeline missed two drains, one of which had a very short intracranial course. In addition, two drains that were in contact with each other were treated as a single object due to the iterative nature of the quantification algorithm, performing the quantification on each discrete object.

2. False positive results (n = 2): A bone fragment incorporated into the ICH and in another patient, a large calcification of the falx cerebri were categorized as a drain.

## Quantification of the drain coverage profile

The highest agreement between predicted and GT coverage profiles in the training and validation groups combined was observed at a threshold of 0.44 (Table 2). At this threshold, the ICC was 0.86 (95% CI: 0.85 to 0.87) in the test dataset. Concordance and Bland-Altman plots (Fig 2 and S3 Fig) showed an average disagreement of -0.005 between the prediction and GT. The number of outliers, in which the disagreement between predicted and GT values was identified outside the 95% confidence interval of the mean difference, was low at 7.69% (305 of 3965) (Table 2).

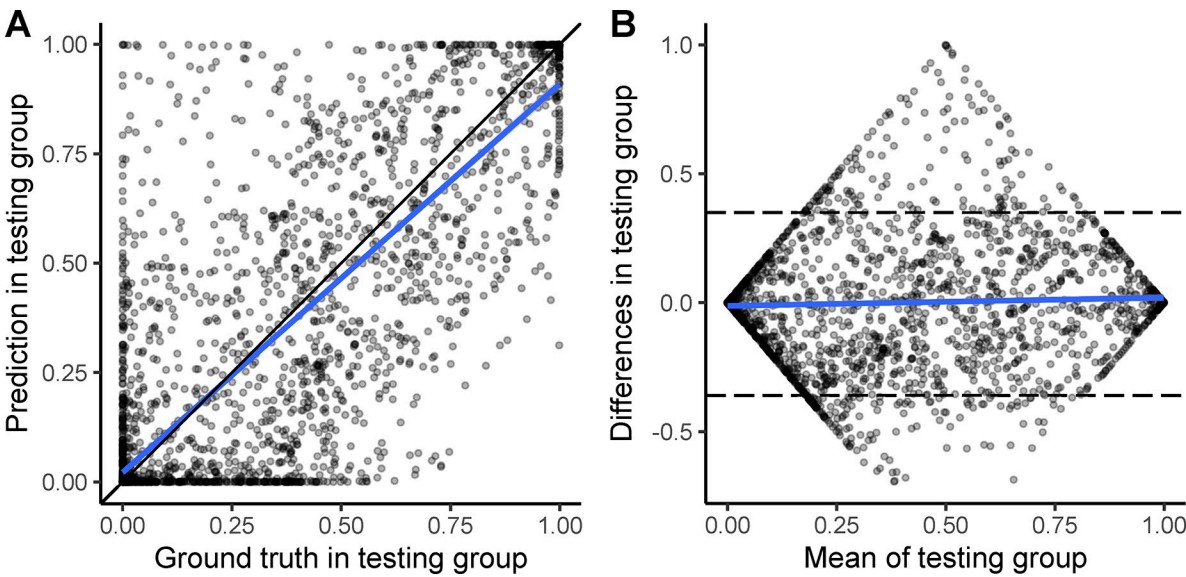

**Fig 2.** Concordance (A) and Bland-Altman plots (B) of the coverage profiles of predicted and GT drains in the testing dataset. Regression line (blue) and 95% confidence interval of predicted values (shaded area).

An illustrative case depicting CT scans before and after surgical correction of an initially misplaced drain is shown in Fig 3.

## Classification of drain position

Given the limited size of the dataset, the results obtained from the combined training and validation datasets are not sufficiently robust to thoroughly assess the model performance. However, they can be utilized to compare between different models. We found no substantial class imbalance in the datasets (Table 2). The best results were observed using penalized logistic regression [15]. The model achieved an AUC-ROC (CI) of 0.98 (95% CI: 0.93 to 1), accuracy (CI) of 0.98 (95% CI: 0.87 to 1), sensitivity (CI) of 0.95 (95% CI: 0.74 to 1) and specificity of one (95% CI: 0.84 to 1). The optimal tuning parameters were a regularization parameter of 6.619512e-04 and a cost-complexity parameter of the logarithm of the sample size.

Table 2 shows the classification results of all correctly detected drains (n = 105), in each group of our data. In the testing dataset we observed an AUC-ROC (S4 Fig) of 0.92 (95% CI: 0.85 to 0.99), accuracy (CI) of 0.88 (57 of 65; 95% CI: 0.77 to 0.95), sensitivity of 0.9 (37 of 41; 95% CI: 0.77 to 0.97) and a specificity of 0.83 (20 of 24; 95% CI: 0.63 to 0.95).

We examined the drains that were inaccurately classified (n = 9, Fig 4). In these drains, the agreement between predicted and GT coverage profiles was moderate to good (ICC = 0.74 [95% CI: 0.7 to 0.77]). Fleiss's κ was -0.07, indicating a poor disagreement between readers, and the human reading was not significantly different from a random classification of the tip position (p = 0.7). In contrast, Fleiss's κ for the 96 correctly classified drains was moderate at 0.63.

1. False negative results (n = 5): In this group, drains #2–4 had intermediate values indicating a borderline drain position. The predicted coverage profiles were generally lower than the GT ones. Drains #7 and #9 had correctly quantified, near-zero coverage profiles, but the human readers classified them as correct. In these cases, misclassification by the human readers was suspected.

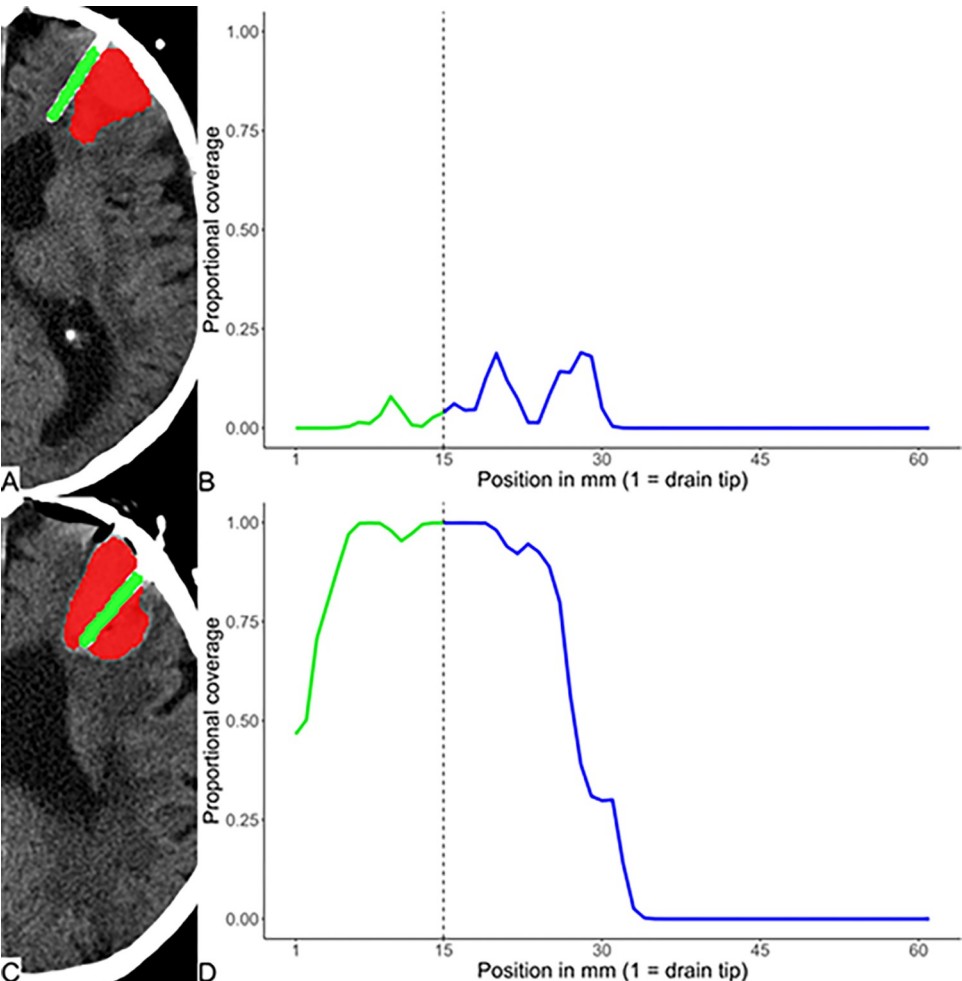

**Fig 3.** Non-contrast axial CT scans (A, C) of a patient from testing dataset following minimally invasive surgery showing a left frontal hemorrhage (red) and drain (green). Corresponding Plots of the coverage profiles (B, D). Vertical dashed line separates the distal 15 mm (green) from the rest of the drain (blue). (A) There is only marginal contact between the drain and the bleeding and (B) a coverage reaching approximately 25% between 15 and 25 mm from the tip. (C) Following surgical correction there was an optimal positioning of the drain and (D) an almost complete coverage of the drain with the hemorrhage.

2. False positive result (n = 4): In this group, drains #1, #5, and #6 showed a possible borderline position of the drain. Drain #8 had a substantially higher predicted profile (Fig 4), leading to a false positive classification. The higher predicted profile was due to an accurate quantification of the coverage of the tip of an external ventricular drain by an intraventricular bleeding (S5 Fig). The drain position was correctly classified by human readers as incorrect because there was no contact with the ICH.

## Discussion

Follow-up CT scans are often performed after minimally invasive drain placement for acute supratentorial ICH and are likely to become increasingly important in light of the ENRICH trial [4]. In our study, we developed an end-to-end pipeline that utilizes a previously published convolutional neural network-based segmentation of ICH and intracranial drains [7] to detect,

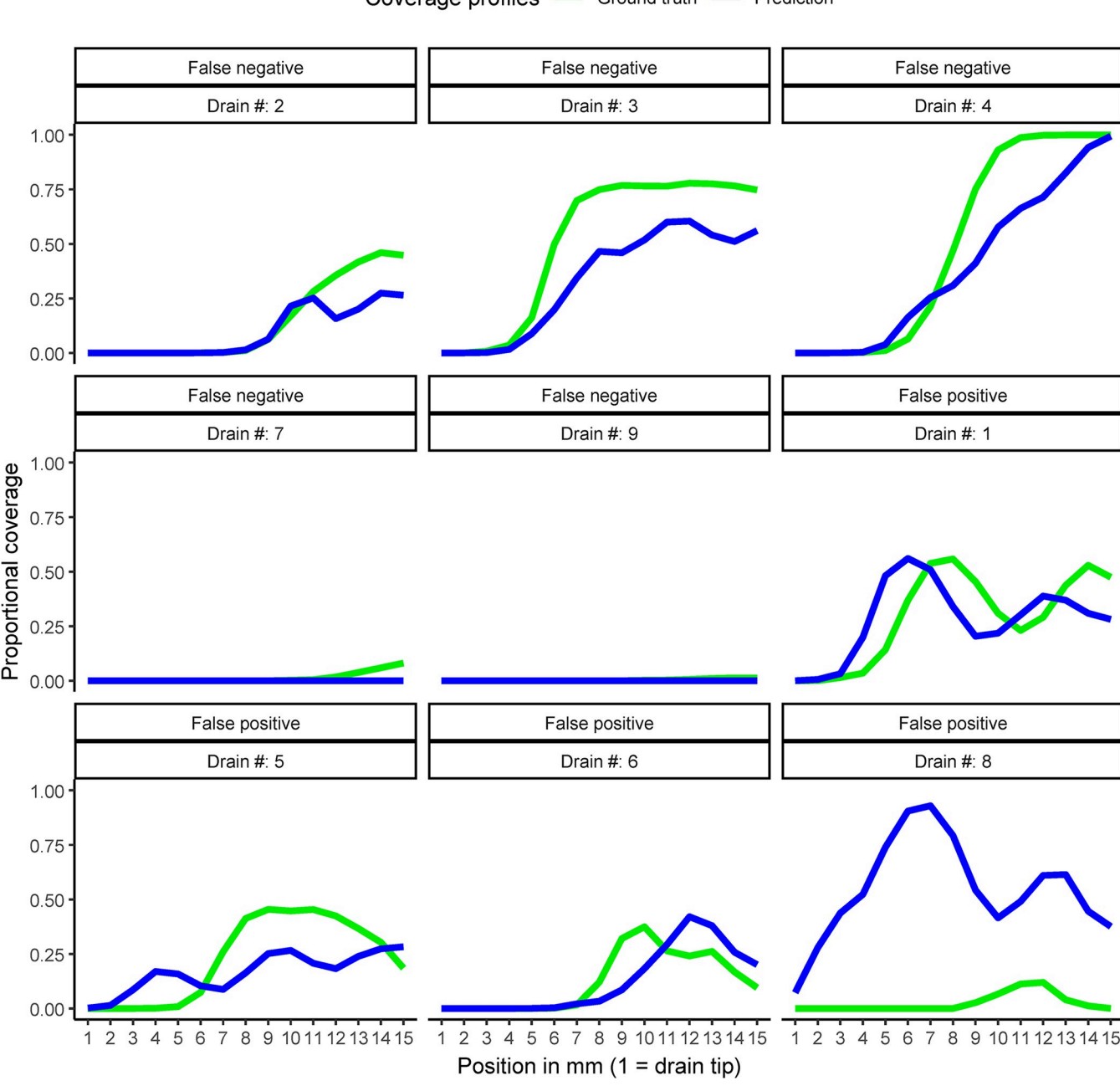

**Fig 4. Figure depicting predicted (blue) and ground truth (green) coverage profiles of drains that were misclassified by the classification model.**

quantify and classify the location of the drain in relation to the ICH. Each step was statistically validated independently, followed by end-to-end testing. The overall drain detection accuracy was excellent at 0.97. We successfully quantified the coverage area between the ICH and the drain reaching an ICC of 0.86 in the test dataset. Using the coverage profiles to classify the tip position, our pipeline achieved a high accuracy of 0.88 (57 of 65; 95% CI: 0.77 to 0.95) with an AUC-ROC of 0.92 (95% CI: 0.85 to 0.99) in identifying incorrect drain positions.

To evaluate generalizability of the model, we included image data originating from different scanners with a wide range of voxel sizes and anisotropy both in the training and the

independent testing datasets. To avoid bias, we tested our pipeline on a dataset of consecutive patients.

The application of machine learning for the detection and segmentation of ICH has been extensively studied, with a detection accuracy of 98% and a dice similarity coefficient for segmentation of 0.92 being achieved [16, 17]. These results are comparable to those obtained with the model used in our pipeline [7]. A review of the literature revealed no publications on the development of segmentation models for ICH drains. Furthermore, our search revealed a paucity of published literature examining the evaluation of drain position, whether using clinical parameters or machine learning approaches. We conducted an expanded search for publications evaluating the placement of shunts in cases of hydrocephalus. Previous studies have employed machine learning to predict shunt failure, but using different methodologies in comparison to our proposed pipeline [18, 19]. These studies utilized either multiple morphological and clinical parameters or only the progression of ventricular volume on serial scans for detection of shunt failure, but no quantification of the coverage was attempted. A technical note demonstrated the successful application of machine learning for the identification of the make and model of ventricular shunts [20].

Due to the absence of published standardized criteria for the evaluation of correct drain position, reporting remains highly subjective. Therefore, even experienced radiologists may classify a borderline position of the drain differently, explaining the moderate overall inter-rater agreement (Fleiss's $\kappa$ = 0.57) in our cohort.

ICH and drains exhibit excellent contrast in CT. This was exploited for automated diagnosis with high accuracy [16, 17]. Elsheikh et. al. [7] observed the best segmentation result in their study using a probability threshold of 0.5 for the binary masks. This however, did not translate to an accurate detection of drains as discrete objects. Using a probability threshold of 0.9, we reached an excellent detection accuracy (0.97 (95% CI: 0.92 to 0.99)) in the test dataset. However, our model showed weaknesses, not detecting a drain with a short intracranial course or in cases with two drains in contact to each other. False positive detections occurred in structures exhibiting high density, e.g. bony fragments and thick calcifications. Enlarging the training cohort could enhance performance in this respect.

The segmentation model results [7] allowed an accurate quantification of the drain coverage, exhibiting a good concordance (ICC = 0.86) with the GT. This facilitated the graphical visualization of the relationship between the drain and the ICH, and served as input for the development of our classification model. In order to address the limitations of the small dataset, we sought to mitigate the issue of overfitting by employing regularized models [21]. Furthermore, a model with a simple architecture is less susceptible to overfitting than a more complex one [21]. We observed this in our results, as the most effective model (penalized logistic regression) was of a relatively simple architecture, which may indicate a certain degree of overfitting with more complex model architectures. Conversely, to enable the model to encompass a sufficient number of data features, we subjected the model to the greatest possible extent of training data by employing leave-one-out cross-validation. We suspect no significant underfitting, as the model reached an accuracy of 0.98 (95% CI: 0.87 to 1) in the training-validation dataset. The model was able to classify the tip position with high accuracy (AUC-ROC = 0.92 (95% CI: 0.85 to 0.99), accuracy = 0.88 [57 of 65; 95% CI: 0.77 to 0.95]) in the testing group. However, in the complete dataset nine drains were incorrectly classified. An examination of the incorrectly classified drains revealed an inherent ambiguity in the drain position, as suggested by the poor interrater agreement (Fleiss' $\kappa$ = -0.07, p = 0.7) in this subset. Although the predicted coverage profiles of these drains demonstrated a moderate to good agreement with GT coverage profiles (ICC = 0.74), this variance may have contributed to the incorrect classifications. The limited dataset size precludes the exclusion of potential inherent

errors in the classification model, as indicated by the somewhat broader 95% confidence intervals, which may also be a contributing factor.

The proposed pipeline has the potential to facilitate numerous research possibilities and clinical applications. Initially, it seems necessary to perform a more comprehensive validation of the prediction model's generalizability using a larger dataset and a more thorough testing of a wider range of machine learning models. Following a more robust validation, the pipeline could be applied for scaling data collection in larger population studies, with the aim of examining the imaging criteria for a correct drain position. In order to achieve the objective of the surgical procedure, which is to reduce ICH volume and improve patient outcome, it would be beneficial to define the imaging criteria associated with these parameters with greater precision. The modular composition of the pipeline allows for straightforward adaptation to a multitude of different tasks. This includes the incorporation of drains with varying working lengths and the application to other disease states using different segmentation models, such as the drainage of subdural hematomas or hydrocephalus. In the forthcoming era of computer-aided diagnosis, further studies could examine whether the graphical representation of the coverage profile and the automated classification would reduce the time needed for the reporting process. This could be accomplished by eliminating the necessity for multiplanar reconstructions and manual measurements. Furthermore, this approach may assist in reducing discrepancies between readers in instances where a borderline positioning of a drain is encountered.

Our study had several limitations. First, the size of the patient cohort was relatively limited, which may have resulted in missing some clinical scenarios that were not encountered in the dataset. However, the issue we sought to examine exhibits limited variability, and our pipeline was accurate when evaluated on a consecutive three-year patient cohort. Additionally, the limited sample size may introduce inherent errors in the classification model and may not allow for adequate measurement of model performance. The model may be susceptible to memorization of the dataset, leading to overfitting, or it may fail to capture sufficient features, leading to underfitting. Such limitations may have a negative impact when the model is applied to previously unseen data. Although the generalizability was tested on a consecutive three-year patient, the total number of included patients and scans was relatively small (n = 44, 59 scans). Second, lacking objective criteria for classifying the drain position, we had to rely on human readers to establish the ground truth for our model. Third, the temporal changes of the ICH after MIS procedures and the relationship with clinical and surgical technical aspects were not investigated in this study, as the primary focus of this study was to establish the diagnostic validity of the pipeline.

## Conclusion

Our proposed, multi-stage, automated end-to-end pipeline to detect drains following minimal invasive surgery in acute ICH and to quantify and then classify the drain position in relation to ICH achieved accurate results, comparable to experienced human readers. Following further validation, the proposed methodology may be applied as a computer-aided diagnostic tool or for data collection in large population studies in different disease states.

## Supporting information

**S1 Fig. Zoomed image of the employed bleeding drain, showing openings only in the distal 15 mm.** Scale is given in cm.
(TIF)

**S2 Fig.** Axial (A-C) and sagittal (D) reconstructions of non-contrast CT scans of falsely detected drains superimposed with the binary drain mask at 0.90 probability threshold (blue). (A) A displaced bone fragment within the bleeding (arrow) and (B) a thick calcification of the flax cerebri (arrow) classified incorrectly as drains. (C) Two touching drains (arrow) segmented as a single object. (D) A drain with a short intracranial course (arrow), missed in the 0.9 threshold binary mask.
(TIF)

**S3 Fig.** Concordance (left column) and Bland-Altman plots (right column) of the coverage profiles of predicted and GT drains in (A) training, (B) validation and (C) testing datasets. Regression line (blue) and 95% confidence interval of predicted values (shaded area).
(TIF)

**S4 Fig. Area under the receiver operating characteristic curve (AUC-ROC) of the drain classification results in testing dataset.** 95% CI: 95% confidence interval.
(TIF)

**S5 Fig.** Image depicting (A) predicted (blue) and ground truth (green) coverage profiles of a false positive classification result (drain #8). (B) Corresponding CT scan, showing contact between tip of the external ventricular drain (blue arrow) and an intraventricular bleeding (red arrow).
(TIF)

**S1 Data. An anonymized, minimal data set to allow replication of the study findings.**
(CSV)

**S2 Data. A data dictionary describing the data type and contents of each column of S1 Data.**
(CSV)

## Author Contributions

**Conceptualization:** Samer Elsheikh, Marco Reisert.

**Data curation:** Samer Elsheikh, Ahmed Elbaz, Alexander Rau, Theo Demerath, Elias Kellner, Ralf Watzlawick, Urs Würtemberger, Horst Urbach, Marco Reisert.

**Formal analysis:** Samer Elsheikh.

**Investigation:** Samer Elsheikh, Marco Reisert.

**Methodology:** Samer Elsheikh, Horst Urbach, Marco Reisert.

**Project administration:** Samer Elsheikh, Marco Reisert.

**Resources:** Elias Kellner, Marco Reisert.

**Software:** Elias Kellner, Marco Reisert.

**Supervision:** Marco Reisert.

**Validation:** Samer Elsheikh.

**Visualization:** Samer Elsheikh.

**Writing – original draft:** Samer Elsheikh, Marco Reisert.

**Writing – review & editing:** Samer Elsheikh, Ahmed Elbaz, Alexander Rau, Theo Demerath, Elias Kellner, Ralf Watzlawick, Urs Würtemberger, Horst Urbach, Marco Reisert.

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
