## [Decision Letter · Decision Letter 0]

7 Nov 2024

PONE-D-24-26212Accuracy of intracerebral hemorrhage drain detection, quantification and classification of coverage in computed tomography using machine learningPLOS ONE

Dear Dr. Elsheikh,

Thank you for submitting your manuscript to PLOS ONE. After careful consideration, we feel that it has merit but does not fully meet PLOS ONE’s publication criteria as it currently stands. Therefore, we invite you to submit a revised version of the manuscript that addresses the points raised during the review process. Please submit your revised manuscript by Dec 22 2024 11:59PM. If you will need more time than this to complete your revisions, please reply to this message or contact the journal office at plosone@plos.org. Please include the following items when submitting your revised manuscript:A rebuttal letter that responds to each point raised by the academic editor and reviewer(s). You should upload this letter as a separate file labeled 'Response to Reviewers'.A marked-up copy of your manuscript that highlights changes made to the original version. You should upload this as a separate file labeled 'Revised Manuscript with Track Changes'.An unmarked version of your revised paper without tracked changes. You should upload this as a separate file labeled 'Manuscript'.

We look forward to receiving your revised manuscript.

Kind regards,

Ziyu Qi

Academic Editor

PLOS ONE

Journal Requirements:

3. Thank you for stating the following in the Competing Interests section: “Samer Elsheikh:

No competing Interests: Unrelated: research grants from Bracco Suisse S.A., Medtronic. Travel grant from Medtronic.

Horst Urbach:

Received honoraria for lectures from Biogen, Eisai, Mbits and Lilly, is supported by German Federal Ministry of Education and Research, and is coeditor of Clin Neuroradiol.

Elias Kellner:

Shareholder of and received fees from VEObrain GmbH, Freiburg, Germany.

Theo Demerath:

No competing interest (unrelated: travel grants Balt, Stryker).”

We note that you received funding from a commercial source: Bracco Suisse S.A., Medtronic, Biogen, Eisai, Mbits and Lilly

Please provide an amended Competing Interests Statement that explicitly states this commercial funder, along with any other relevant declarations relating to employment, consultancy, patents, products in development, marketed products, etc. Within this Competing Interests Statement, please confirm that this does not alter your adherence to all PLOS ONE policies on sharing data and materials by including the following statement: "This does not alter our adherence to PLOS ONE policies on sharing data and materials.” (as detailed online in our guide for authors http://journals.plos.org/plosone/s/competing-interests). If there are restrictions on sharing of data and/or materials, please state these. Please note that we cannot proceed with consideration of your article until this information has been declared. Please include your amended Competing Interests Statement within your cover letter. We will change the online submission form on your behalf.

4. We note that you have indicated that there are restrictions to data sharing for this study. PLOS only allows data to be available upon request if there are legal or ethical restrictions on sharing data publicly. For more information on unacceptable data access restrictions, please see http://journals.plos.org/plosone/s/data-availability#loc-unacceptable-data-access-restrictions. Before we proceed with your manuscript, please address the following prompts: a) If there are ethical or legal restrictions on sharing a de-identified data set, please explain them in detail (e.g., data contain potentially identifying or sensitive patient information, data are owned by a third-party organization, etc.) and who has imposed them (e.g., a Research Ethics Committee or Institutional Review Board, etc.). Please also provide contact information for a data access committee, ethics committee, or other institutional body to which data requests may be sent. b) If there are no restrictions, please upload the minimal anonymized data set necessary to replicate your study findings to a stable, public repository and provide us with the relevant URLs, DOIs, or accession numbers. For a list of recommended repositories, please see https://journals.plos.org/plosone/s/recommended-repositories. You also have the option of uploading the data as Supporting Information files, but we would recommend depositing data directly to a data repository if possible. We will update your Data Availability statement on your behalf to reflect the information you provide.

6. We are unable to open your Supporting Information file “S3_Fig.eps”. Please kindly revise as necessary and re-upload.

Additional Editor Comments:

Before we consider accepting this manuscript for publication, you must fully address the concerns raised by Reviewers 1 and 2.

Reviewers' comments:

Reviewer's Responses to Questions

**Comments to the Author**

1. Is the manuscript technically sound, and do the data support the conclusions?

Reviewer #1: Yes

Reviewer #2: Yes

2. Has the statistical analysis been performed appropriately and rigorously? 

Reviewer #1: Yes

Reviewer #2: Yes

3. Have the authors made all data underlying the findings in their manuscript fully available?

Reviewer #1: Yes

Reviewer #2: No

4. Is the manuscript presented in an intelligible fashion and written in standard English?

Reviewer #1: Yes

Reviewer #2: Yes

5. Review Comments to the Author

Reviewer #1: The manuscript presents a promising approach to the automated detection of intracerebral hemorrhage using convolutional neural networks. It effectively addresses a significant clinical need by proposing a novel detection pipeline and demonstrating its potential efficacy. However, the study is limited by the small dataset size, which may affect the robustness and generalizability of the findings. While the results are promising, the manuscript would benefit from a more detailed discussion of the dataset limitations and their impact on the results. Additionally, enhancements to the title and abstract are needed to better reflect the study's focus and key findings. Overall, the research provides valuable insights but requires further validation and refinement. Please see below for my detailed comments.

Title:

• The title should clearly indicate the focus of the study, specifically emphasizing the automated nature of the pipeline and its application to ICH detection.

• Aim for a concise title that effectively conveys the essence of the study without being overly complex.

Abstract:

• Clearly state the main aim of the study, such as developing and validating an automated pipeline for ICH detection from imaging data. Provide context on the importance of the research and what gaps it addresses.

• Discuss the implications of the findings and their significance for clinical practice or future research. Highlight the contribution of the study to the field.

Introduction:

• Context and Background: The introduction provides a good overview of intracerebral hemorrhage and its clinical significance. However, it could benefit from a more detailed discussion of current detection methods and their limitations to set up the research gap more clearly.

• Research Gap and Objectives: The manuscript should explicitly state the research gap it addresses and the specific objectives of the study. This will help in aligning the reader’s understanding with the study’s contributions.

Methods:

• Data Description: The manuscript lacks detail on the dataset size and composition. Given the small data size, a more thorough description of the dataset, including the number of images, sources, and any potential biases, is crucial.

• Model Architecture: More detail is needed on the convolutional neural network architecture used, including the number of layers, types of layers, and any hyperparameters that were tuned.

• Validation Strategy: The method of model validation should be described in greater detail. Explain if cross-validation, a holdout set, or other techniques were used, and how they contribute to the robustness of the results.

• Offer a clearer explanation of the convolutional neural network architecture used. Include details on the layers, activation functions, and any hyperparameters.

• Elaborate on the validation process, including any cross-validation techniques used. Specify how performance metrics were calculated and provide justification for the chosen methods.

• The limited size of the dataset is a critical concern. Discuss how the small data size might impact the training and generalization of the model.

Results:

• Reflect on how the small dataset size might influence the performance metrics and results. Discuss any potential overfitting issues and how they were addressed. Provide insights into how the model's performance might vary with a larger dataset.

Discussion:

• Interpretation of Results: The discussion should offer a deeper interpretation of the results, including how the performance metrics compare with those from previous studies.

• Limitations: There should be a more explicit discussion of the study’s limitations, especially concerning the small dataset size. Address how these limitations might affect the generalizability and reliability of the results.

• Future Work: Suggestions for future research should be more detailed, including how to address the limitations and potential improvements in dataset size or model complexity.

Conclusion:

• Summary: Summarize the main findings of the study and their significance. Emphasize the contribution of the research to the field of ICH detection and any practical implications for clinical practice.

• Implications: Clearly outline the potential impact of the study on current practices and future research directions.

Reviewer #2: This study applies machine learning techniques to the detection of drain in CT scan images of intracranial hemorrhages after minimally invasive intervention, as well as to the quantification and classification of drainage coverage.

Article well structured and written. Methodology is correct.

The number of patients included in the training and validation subgroup is exceedingly small (21 and 3, respectively). A similar situation is observed with the testing group, which initially comprised only 5 patients but was subsequently expanded to 44 in a second phase. This expansion allows for a more reliable evaluation of the algorithm's validity in clinical practice.

It may seem surprising that, with such a limited number of subjects in the training set, acceptable accuracy values are achieved for detection (97%), coverage (86%), and correct catheter positioning (88%). I believe this is an example where the task is relatively straightforward for an algorithm, and the errors exhibited could be minimized by increasing the number of training cases. The authors acknowledge the significant limitations of their study in the discussion section. Despite these limitations, I consider this to be a well-designed study with useful results, particularly in the context of a relatively uncommon pathology for which it may be challenging to obtain a larger patient cohort.

Minor Comments

- I recommend avoiding the use of abbreviations in the abstract.

6. PLOS authors have the option to publish the peer review history of their article (what does this mean?). If published, this will include your full peer review and any attached files.

Reviewer #1: No

Reviewer #2: **Yes: **Juan-Jose Beunza

---

## [Author Response · Author response to Decision Letter 0]

26 Nov 2024

The responses to the editorial offfice concerning formatting and revisions of the ethics statement and data availabliity are included point for point in the new covering letter. Please find below the point for point responses to the reviewers. 

Point for Point Response

Reviewer 1:

The manuscript presents a promising approach to the automated detection of intracerebral hemorrhage using convolutional neural networks. It effectively addresses a significant clinical need by proposing a novel detection pipeline and demonstrating its potential efficacy. However, the study is limited by the small dataset size, which may affect the robustness and generalizability of the findings. While the results are promising, the manuscript would benefit from a more detailed discussion of the dataset limitations and their impact on the results. Additionally, enhancements to the title and abstract are needed to better reflect the study’s focus and key findings. Overall, the research provides valuable insights but requires further validation and refinement. Please see below for my detailed comments.

Response

We thank you for your valuable comments and input. The comments address important points, some were omitted and some were reported too concisely. We made changes to the manuscript, which we believe to have addressed all the points in the comments.

Below are point for point responses, highlighting the changes we made, below each comment.

Title:

• The title should clearly indicate the focus of the study, specifically emphasizing the automated nature of the pipeline and its application to ICH detection. 

• Aim for a concise title that effectively conveys the essence of the study without being overly complex.

Response

Original title: Accuracy of intracerebral hemorrhage drain detection, quantification and classification of coverage in computed tomography using machine learning

We changed the Title to: “Machine Learning-Based Pipeline for Automated Intracerebral Hemorrhage and Drain Detection, Quantification, and Classification in non-enhanced CT Images (NeuroDrAIn)”.

Abstract:

• Clearly state the main aim of the study, such as developing and validating an automated pipeline for ICH detection from imaging data. Provide context on the importance of the research and what gaps it addresses. 

• Discuss the implications of the findings and their significance for clinical practice or future research. Highlight the contribution of the study to the field.

Response

Original: Background and purpose To develop and validate an automated pipeline for intracerebral hemorrhage (ICH) drain detection, quantification of ICH coverage and detection of malpositioned drains.

Changed to: Background and purpose External drainage represents a well-established treatment option for acute intracerebral hemorrhage. The current standard of practice includes post-operative CT imaging, which is subjectively evaluated. The implementation of an objective, automated evaluation of postoperative studies would enhance diagnostic accuracy and facilitate the scaling of research projects. The objective is to develop and validate a fully automated pipeline for intracerebral hemorrhage and drain detection, quantification of intracerebral hemorrhage coverage, and detection of malpositioned drains.

Introduction:

• Context and Background: The introduction provides a good overview of intracerebral hemorrhage and its clinical significance. However, it could benefit from a more detailed discussion of current detection methods and their limitations to set up the research gap more clearly.

Response

We added the following sentences:

Moreover, the lack of research on standardized or clinically validated criteria for detecting malpositioned drains represents a significant limitation to both clinical and scientific evaluation. Consequently, the assessment of these images remains highly subjective.

• Research Gap and Objectives: The manuscript should explicitly state the research gap it addresses and the specific objectives of the study. This will help in aligning the reader’s understanding with the study’s contributions.

Response

We added the following:

The application of automated segmentation techniques using convolutional neural networks to medical images provides the potential for the generation of quantitative data. If validated, this data could facilitate objective clinical assessment and serve as a foundation for further scientific research.

Methods:

• Data Description: The manuscript lacks detail on the dataset size and composition. Given the small data size, a more thorough description of the dataset, including the number of images, sources, and any potential biases, is crucial.

Response

To improve the presentation of information, we added the number of images belonging to each group to the methods sections. We explicitly pointed out the steps we have taken to reduce bias.

• Model Architecture: More detail is needed on the convolutional neural network architecture used, including the number of layers, types of layers, and any hyperparameters that were tuned.

• Validation Strategy: The method of model validation should be described in greater detail. Explain if cross-validation, a holdout set, or other techniques were used, and how they contribute to the robustness of the results.

• Offer a clearer explanation of the convolutional neural network architecture used. Include details on the layers, activation functions, and any hyperparameters.

Response

Instead of referring the reader to the previously published work we added a 1-paragraph summary describing the model architecture, the development and tuning process, the validation strategy and the rationale behind it and the evaluation metrics. We also highlighted the most relevant ones.

• Elaborate on the validation process, including any cross-validation techniques used. Specify how performance metrics were calculated and provide justification for the chosen methods.

• The limited size of the dataset is a critical concern. Discuss how the small data size might impact the training and generalization of the model.

Response

This was a concern while performing the study. We deemed it important to maintain the same data partitioning as that used in developing the CNN to avoid data leakage, so we decided against a repartitioning of the images.

We added this paragraph:

The small size of our data set may present challenges in developing a predictive classification model. To prevent overfitting we employed regularized models, but did not exclude any of the more complex model families. To limit bias we utilized leave-one-out cross validation, which would maximize the amount of data during training. Finally, we tested model robustness on an independent testing dataset. As a thorough evaluation of the performance could prove difficult, we included the 95%-confidence intervals in all our results. Still we recognize that a thorough evaluation of model performance will be difficult.

Results:

• Reflect on how the small dataset size might influence the performance metrics and results. Discuss any potential overfitting issues and how they were addressed. Provide insights into how the model’s performance might vary with a larger dataset.

Response

We added this sentence:

Given the limited size of the dataset, the results obtained from the combined training and validation datasets are not sufficiently robust to thoroughly assess the model performance. However, they can be utilized to compare between different models.

We further elaborated on this issue in the discussions section.

Discussion:

• Interpretation of Results: The discussion should offer a deeper interpretation of the results, including how the performance metrics compare with those from previous studies.

Response

We repeated the literature search, and still found no methodologically comparable studies, so we expanded our search for studies, that may have tackled a similar problem in other disease states or body regions. We added this paragraph:

The application of machine learning for the detection and segmentation of ICH has been extensively studied, with a detection accuracy of 98% and a dice similarity coefficient for segmentation of 0.92 being achieved [16, 17]. These results are comparable to those obtained with the model used in our pipeline [7]. A review of the literature revealed no publications on the development of segmentation models for ICH drains. Furthermore, our search revealed a paucity of published literature examining the evaluation of drain position, whether using clinical parameters or machine learning approaches. We conducted an expanded search for publications evaluating the placement of shunts in cases of hydrocephalus. Previous studies have employed machine learning to predict shunt failure, but using different methodologies in comparison to our proposed pipeline [18, 19]. These studies utilized either multiple morphological and clinical parameters or only the progression of ventricular volume on serial scans for detection of shunt failure, but no quantification of the coverage was attempted. A technical note demonstrated the successful application of machine learning for the identification of the make and model of ventricular shunts [20].

Response

In response to the above mentioned point concerning reflection on the effect of data size on the results, we changed an existing paragraph to read as follows:

The segmentation model results [7] allowed an accurate quantification of the drain coverage, exhibiting a good concordance (ICC = 0.86) with the GT. This facilitated the graphical visualization of the relationship between the drain and the ICH, and served as input for the development of our classification model. In order to address the limitations of the small dataset, we sought to mitigate the issue of overfitting by employing regularized models [21]. Furthermore, a model with a simple architecture is less susceptible to overfitting than a more complex one [21]. We observed this in our results, as the most effective model (penalized logistic regression) was of a relatively simple architecture, which may indicate a certain degree of overfitting with more complex model architectures. Conversely, to enable the model to encompass a sufficient number of data features, we subjected the model to the greatest possible extent of training data by employing leave-one-out cross-validation. We suspect no significant underfitting, as the model reached an accuracy of 0.98 (95% CI: 0.87 to 1) in the training-validation dataset. The model was able to classify the tip position with high accuracy (AUC-ROC = 0.92 (95% CI: 0.85 to 0.99), accuracy = 0.88 [57 of 65; 95% CI: 0.77 to 0.95]) in the testing group. However, in the complete dataset nine drains drains were incorrectly classified. An examination of the incorrectly classified drains revealed an inherent ambiguity in the drain position, as suggested by the poor interrater agreement (Fleiss’ κ = -0.07, p = 0.7) in this subset. Although the predicted coverage profiles of these drains demonstrated a moderate to good agreement with GT coverage profiles (ICC = 0.74), this variance may have contributed to the incorrect classifications. The limited dataset size precludes the exclusion of potential inherent errors in the classification model, as indicated by the somewhat broader 95%-confidence intervals, which may also be a contributing factor.

• Limitations: There should be a more explicit discussion of the study’s limitations, especially concerning the small dataset size. Address how these limitations might affect the generalizability and reliability of the results.

Response

We added the following to the pre-existing paragraph:

Our study had several limitations. First, the size of the patient cohort was relatively limited, which may have resulted in missing some clinical scenarios that were not encountered in the dataset. However, the issue we sought to examine exhibits limited variability, and our pipeline was accurate when evaluated on a consecutive three-year patient cohort. Additionally, the limited sample size may introduce inherent errors in the classification model and may not allow for adequate measurement of model performance. The model may be susceptible to memorization of the dataset, leading to overfitting, or it may fail to capture sufficient features, leading to underfitting. Such limitations may have a negative impact when the model is applied to previously unseen data. Although the generalizability was tested on a consecutive three-year patient, the total number of included patients and scans was relatively small (n = 44, 59 scans).

• Future Work: Suggestions for future research should be more detailed, including how to address the limitations and potential improvements in dataset size or model complexity.

Response

we changed an existing paragraph to read as follows:

The proposed pipeline has the potential to facilitate numerous research possibilities and clinical applications. Initially, it seems necessary to perform a more comprehensive validation of the prediction model’s generalizability using a larger dataset and a more thorough testing of a wider range of machine learning models. Following a more robust validation, the pipeline could be applied for scaling data collection in larger population studies, with the aim of examining the imaging criteria for a correct drain position. In order to achieve the objective of the surgical procedure, which is to reduce ICH volume and improve patient outcome, it would be beneficial to define the imaging criteria associated with these parameters with greater precision. The modular composition of the pipeline allows for straightforward adaptation to a multitude of different tasks. This includes the incorporation of drains with varying working lengths and the application to other disease states using different segmentation models, such as the drainage of subdural hematomas or hydrocephalus. In the forthcoming era of computer-aided diagnosis, further studies could examine whether the graphical representation of the coverage profile and the automated classification would reduce the time needed for the reporting process. This could be accomplished by eliminating the necessity for multiplanar reconstructions and manual measurements. Furthermore, this approach may assist in reducing discrepancies between readers in instances where a borderline positioning of a drain is encountered.

Conclusion: 

• Summary: Summarize the main findings of the study and their significance. Emphasize the contribution of the research to the field of ICH detection and any practical implications for clinical practice. 

• Implications: Clearly outline the potential impact of the study on current practices and future research directions.

Response

We expanded the conclusion in the manuscript and in the abstract to include theese points:

Our proposed, multi-stage, automated end-to-end pipeline to detect drains following minimal invasive surgery in acute ICH and to quantify and then classify the drain position in relation to ICH achieved accurate results, comparable to experienced human readers. Following further validation, the proposed methodology may be applied as a computer-aided diagnostic tool or for data collection in large population studies in different disease states.

Reviewer #2

This study applies machine learning techniques to the detection of drain in CT scan images of intracranial hemorrhages after minimally invasive intervention, as well as to the quantification and classification of drainage coverage. Article well structured and written. Methodology is correct. The number of patients included in the training and validation subgroup is exceedingly small (21 and 3, respectively). A similar situation is observed with the testing group, which initially comprised only 5 patients but was subsequently expanded to 44 in a second phase. This expansion allows for a more reliable evaluation of the algorithm’s validity in clinical practice. It may seem surprising that, with such a limited number of subjects in the training set, acceptable accuracy values are achieved for detection (97%), coverage (86%), and correct catheter positioning (88%). I believe this is an example where the task is relatively s

---

## [Editor Report · Decision Letter 1]

5 Dec 2024

Machine Learning-Based Pipeline for Automated Intracerebral Hemorrhage and Drain Detection, Quantification, and Classification in Non-Enhanced CT Images (NeuroDrAIn)

PONE-D-24-26212R1

Dear Dr. Elsheikh,

We’re pleased to inform you that your manuscript has been judged scientifically suitable for publication and will be formally accepted for publication once it meets all outstanding technical requirements.

Kind regards,

Ziyu Qi

Academic Editor

PLOS ONE

Additional Editor Comments (optional):

The manuscript can be accepted for publication in its present form. Congrats!
---

## [Editor Report · Acceptance letter]

12 Dec 2024

PONE-D-24-26212R1 

PLOS ONE

Dear Dr. Elsheikh, 

I'm pleased to inform you that your manuscript has been deemed suitable for publication in PLOS ONE. Congratulations! Your manuscript is now being handed over to our production team.

Kind regards, 

on behalf of

Mr. Ziyu Qi 

Academic Editor

PLOS ONE